# Enantioselective alkylative cross-coupling of unactivated aromatic C–O electrophiles

Zishuo Zhang[1,4], Jintong Zhang[1,4], Quan Gao[1,4], Yu Zhou[2], Mingyu Yang[3], Haiqun Cao[1], Tingting Sun[1], Gen Luo [2✉] & Zhi-Chao Cao [1✉]

Nonpolar alkyl moieties, especially methyl group, are frequently used to modify bioactive molecules during lead optimization in medicinal chemistry. Thus transition-metal catalyzed alkylative cross-coupling reactions by using readily available and environmentally benign C–O electrophiles have been established as powerful tools to install alkyl groups, however, the C(sp³)–C(sp²) cross-coupling via asymmetric activation of aromatic C–O bond for the synthesis of alkylated chiral compounds remains elusive. Here, we unlock a C(sp³)–C(sp²) cross-coupling via enantioselective activation of aromatic C–O bond for the efficient synthesis of versatile axially chiral 2-alkyl-2'-hydroxyl-biaryl compounds. By employing a unique chiral N-heterocyclic carbene ligand, this transformation is accomplished via nickel catalysis with good enantiocontrol. Mechanistic studies indicate that bis-ligated nickel complexes might be formed as catalytically active species in the enantioselective alkylative cross-coupling. Moreover, further derivation experiments suggest this developed methodology holds great promise for complex molecule synthesis and asymmetric catalysis.

¹ Anhui Province Engineering Laboratory for Green Pesticide Development and Application, College of Plant Protection, Anhui Agricultural University, Hefei, Anhui 230036, China. ² Institute of Physical Science and Information Technology, Anhui University, Hefei 230601, China. ³ School of Chemistry & Chemical Engineering, Shaanxi Normal University, Xi'an, Shaanxi 710119, China. ⁴These authors contributed equally: Zishuo Zhang, Jintong Zhang, Quan Gao. ✉email: luogen@ahu.edu.cn; zc_cao@ahau.edu.cn

Incorporation of nonpolar alkyl moieties, especially methyl group, have been established as a powerful tool to modify bioactive molecules during lead optimization in medicinal chemistry[1,2]. For example, the methylated biphenyl amides (BPAs) exhibits a 200-fold increase of binding affinity ($K_i$) of p38α MAP kinase than the original BAPs (Fig. 1a)[3]. Thus developing strategies for the efficient and direct incorporation of nonpolar alkyl groups represents an attractive goal in organic synthesis. Trigged by the easy availability and natural abundance of oxygen-based compounds, transition-metal catalyzed C(sp³)–C(sp²) cross-coupling reactions via aromatic C–O bond activation received many attentions during the past decades, and have been developed as a powerful tool for the installation of alkyl groups (Fig. 1b)[4–10]. For example, pioneered by Wenkert's work in 1984[11], a wide range of efficient protocols for alkylative cleavage of aromatic C–O bond has been established[12–17]. Rueping and co-workers also reported an efficient dealkoxylative

alkylation of aryl ethers by employing a lithium bifunctional nucleophile[18]. Moreover, the groups of Chatani, Tobisu, and Rueping disclosed more general C(sp³)–C(sp²) cross-couplings via C–O bond activation[19–22]. Very recently, Shi and co-workers demonstrated the nickel-catalyzed methylative cross-coupling by using arenols as the starting material directly[23]. Despite these advances, no examples on C(sp³)–C(sp²) cross-coupling via asymmetric aromatic C–O bond activation have been reported.

On the other hand, axially chiral biaryl scaffolds, including the axially chiral compounds of type **A** (2-alkyl-2'-hydroxyl-biaryls), are found widespread in natural products[24–26], and most of them have been identified as bioactive molecules[27–30]. Of particular note, in asymmetric synthesis, a myriad of privileged ligands and catalysts can be easily derived from axially chiral 2-methyl-2'-hydroxyl-biaryl structure motif (Fig. 1c)[31–38]. Although a huge amount of efforts has been directed toward the efficient synthesis of axially chiral biaryls, and various strategies were established[39–43], the

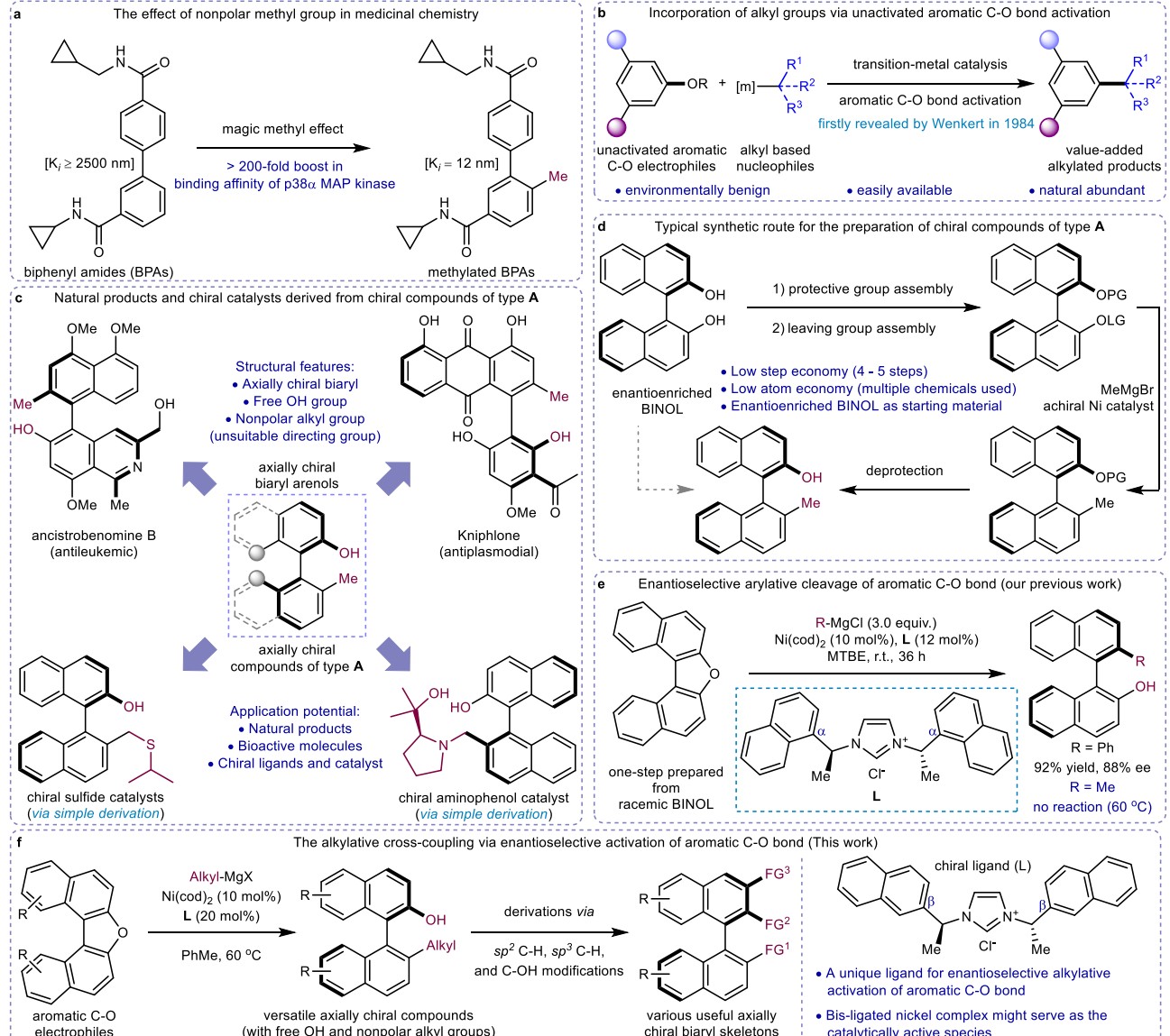

**Fig. 1 Enantioselective C(sp³)–C(sp²) cross-coupling of unactivated aromatic C–O bond. a** The effect of nonpolar methyl group in medicinal chemistry. **b** Incorporation of alkyl groups via unactivated aromatic C–O bond activation. **c** Natural products and chiral catalysts derived from chiral compounds of type **A**. **d** Typical synthetic route for the preparation of chiral compounds of type **A**. **e** Enantioselective arylative cleavage of aromatic C–O bond (our previous work). **f** This work: the alkylative cross-coupling via enantioselective activation of aromatic C–O bond. BINOL 1,1'-bi-2-naphthol, PG protective group, LG leaving group, FG functional group, MTBE methyl tertbutyl ether.

synthesis of axially chiral scaffolds of type **A** remains a challenge, likely due to the incapacity of nonpolar alkyl groups as a directing group and the poor compatibility with free ortho-hydroxyl functional group (OH) in these methodologies. Conventionally, the preparation of axially chiral compounds of type **A** was realized via multiple steps, including protection of hydroxyl group, leaving group assembly, transition-metal catalyzed cross-coupling, and deprotection, by using enantioenriched 1,1′-bi-2-naphthol (BINOL) derivatives as the starting materials, thus the practical applications of this axially chiral biaryl scaffold in asymmetric synthesis and drug discovery were limited due to the poor step- and atom-economy (Fig. 1d)[44–46].

Recently, transition-metal catalyzed enantioselective ring-opening cross-coupling reaction is emerging as a promising strategy for the synthesis of axially chiral biaryl skeletons that possessing native functional groups[47–56]. By employing this strategy, we wondered whether an enantioselective $C(sp^3)$–$C(sp^2)$ cross-coupling reaction could be disclosed via transition-metal catalyzed asymmetric aromatic C–O bond cleavage of diarylfuran derivatives. Such an enantioselective catalytic protocol will possess huge synthetic potentials to access more diverse functionalized axially chiral 2-alkyl-2′-hydroxyl-biaryl scaffolds, thereby providing a highly desirable solution to address the aforementioned challenges in asymmetric catalysis. However, enantioselective ring-opening reactions involving $C(sp^3)$–$C(sp^2)$ bond formation have been rarely reported at current stage, likely as a result of the high barrier of reductive elimination from transition-metal complexes[57–59]. For example, to date, only Hayashi and co-workers have reported an enantioselective methylative ring-opening cross-coupling by using dinaphthylthiophene as the starting material, and only 68% ee was observed[51]. Moreover, in our previously reported enantioselective arylative cross-coupling, methyl Grignard reagent was demonstrated as unsuitable nucleophile, and no desired product was delivered, further suggested the challenges in the designed enantioselective $C(sp^3)$–$C(sp^2)$ cross-coupling of unactivated aromatic C–O electrophiles (Fig. 1e)[54].

In this work, by employing a unique chiral N-heterocyclic carbene (NHC) ligand, we overcome the challenges and unlock the $C(sp^3)$–$C(sp^2)$ cross-coupling protocol via enantioselective aromatic C–O bond activation (Fig. 1f). Experimental and computational studies of mechanism are conducted and suggest that *bis*-ligated nickel complexes might serve as the catalytically active species for this enantioselective alkylative cross-coupling of aromatic C–O bond. By using the developed catalytic system, versatile axially chiral 2-alkyl-2′-hydroxyl-biaryl skeletons are delivered in high yields and with high enantioselectivity (up to 99% yield and 99.5% ee), and further derivations are conducted for the synthesis of various axially chiral biaryl skeletons.

## Results and discussion

**Reaction development and optimization.** We began our investigation by choosing the cross-coupling of dinaphthylfuran **1a** with methyl magnesium bromide **2a** as the model system for reaction conditions optimization due to the magic methyl effect in drug discovery (Table 1)[1]. After examination of various reaction parameters, we were delighted to find that a combination of Ni(cod)$_2$ (10 mol%) and chiral NHC ligand **L1** (20 mol%) could facilitate the desired enantioselective cross-coupling efficiently in toluene (0.1 M) at 60 °C for 24 h, and the desired axially chiral biaryl arenol **3a** could be obtained in 97% isolated yield and with 98% ee (entry 1). The control experiments revealed the pivotal roles of catalyst and ligand in this transformation (entries 2–3). For example, no desire product was observed in the absence of nickel catalyst. On the basis of these results, other nickel

precatalysts, such as air-stable NiBr$_2$(dme) and NiBr$_2$, were used as alternatives in the designed enantioselective cross-coupling reaction (entries 4–5). While comparable results could be observed when NiBr$_2$(dme) was used, the compound **3a** was obtained in 18% NMR yield and 78% ee when NiBr$_2$ (10 mol%) was used as the catalyst, maybe as a result of the poor solubility in toluene.

Next, we evaluated other reaction dimensions using Ni(cod)$_2$ as the precatalyst. As ligand plays a pivotal role in the nickel-catalyzed enantioselective cross-coupling, the effect of ligands was carefully investigated, and the results demonstrated the unique role of ligand **L1** to impact the enantiocontrol in the designed enantioselective cross-coupling (entry 6). In stark contrast, ligand **L2**, which proved to be the optimal ligand in our previous work failed to facilitate the enantioselective $C(sp^3)$–$C(sp^2)$ cross-coupling, and only trace amount of **3a** were observed. The performances of other chiral NHC ligands (**L3**–**L8**) were also evaluated, but no improved results were obtained. For example, using ligands **L3**–**L7**, produced compound **3a** in nearly quantitative yields, but poor enantioselectivities were observed. Solvent screening indicated that toluene was superior to polar solvents (entries 7–10). For example, the cross-coupling led to low yields (<10%) when DCM and dioxane were used as the solvent. The enantioselective alkylative cross-coupling was also conducted at different temperatures (entries 11–12), and comparable results could be obtained. While a slightly diminished yield and enantioselectivity were obtained when 12 mol% chiral NHC ligand **L1** or a half of the standard catalyst loading was used (entries 13–14), only trace amount of product **3a** was observed when 1 mol% nickel catalyst was used (entry 15). Interestingly, methyl magnesium iodide provided compound **3a** in low yield but with good ee, whereas methyl magnesium chloride afforded compound **3a** in high yield with modest ee (entries 16–17), these results clearly indicated the pivotal ionic effect in this enantioselective cross-coupling.

**Evaluation of substrate scope.** Having developed suitable reaction conditions for the enantioselective alkylative cross-coupling, we next embarked to examine the substrate scope of the nickel-catalyzed enantioselective $C(sp^3)$–$C(sp^2)$ cross-coupling via aromatic C–O bond activation. As illustrated in Fig. 2, the C–O electrophiles bearing alkyl groups at 6- and 6'-positions were identified as suitable substrates, and consistently high yields and ee were observed (**3a**–**3e**). However, in the case of an electrophile bearing secondary alkyl group, such as cyclo-hexyl group, the targeted product **3f** was obtained in moderate yield and with moderate ee. The chiral arenols (**3g**–**3h**) could be furnished axially with satisfied results when alkenyl and phenyl groups were introduced into substrates. To our delight, a variety of functional groups, including amine (**3i**), ether (**3j**), and trifluoromethyl group (**3k**), were compatible with this enantioselective $C(sp^3)$–$C(sp^2)$ cross-coupling protocol. With regard to the position effect of substituents in C–O electrophiles, the substrates bearing substituents at 7- and 7′-postions were prepared and examined, and the desired products **3l** and **3m** were forged in high yields and with high ee. Additionally, the substrate bearing substituents at 4- and 4′-postion was also subjected to the catalytic system, the axially chiral arenol **3n** was obtained in 81% yield and with 94% ee.

Turning to the nucleophile scope (Fig. 2), we found that the enantioselective cross-coupling of C–O electrophile **1a** with arylmethyl Grignard reagents could proceed smoothly and delivered the corresponding products (**3o**–**3aa**) efficiently. For example, the nucleophiles that possess alkyl substituents at different positions could deliver the corresponding products in high yields with high ee (**3o**–**3r**). The substituents in Grignard reagents can vary in size from methyl group to iso-propyl group and tert-

**Table 1 Effect of reaction parameters.**

| Entry | Variation from standard conditions | Yield (%) | ee (%) |
|---|---|---|---|
| 1 | none | 96(97) | 98 |
| 2 | no Ni(cod)$_2$ | 0 | n.d. |
| 3 | no **L1** | <5 | n.d. |
| 4 | NiBr$_2$(dme) instead of Ni(cod)$_2$ | 97 | 97 |
| 5 | NiBr$_2$ instead of Ni(cod)$_2$ | 18 | 78 |
| 6 | **L2-L8** instead of **L1** | listed as below | |
| 7 | THF instead of PhMe | 68 | 91 |
| 8 | DCM instead of PhMe | <5 | n.d. |
| 9 | 1,4-dioxane instead of PhMe | 9 | 58 |
| 10 | DME instead of PhMe | 56 | 72 |
| 11 | 45 °C instead of 60 °C | 95 | 98 |
| 12 | 80 °C instead of 60 °C | 96 | 95 |
| 13 | Ni(cod)$_2$ (10 mol%), **L1** (12 mol%) | 96 | 94 |
| 14 | Ni(cod)$_2$ (5 mol%), **L1** (10 mol%) | 95 | 94 |
| 15 | Ni(cod)$_2$ (1 mol%), **L1** (2 mol%) | <5 | n.d. |
| 16 | Me-MgI instead of Me-MgBr | 25 | 94 |
| 17 | Me-MgCl instead of Me-MgBr | 95 | 70 |

The reactions were carried out by C–O electrophile **1a** (0.1 mmol), methyl Grignard reagent (3.0 equiv.), Ni(cod)$_2$ (10 mol%), chiral ligand (20 mol%) in solvent (0.1 M) at indicated temperature for 24 h; $^1$H-NMR yields were reported by using CH$_2$Br$_2$ as internal standard, isolated yield in parentheses was reported; ee were determined by using chiral high-performance liquid chromatography (HPLC) analysis.

ee enantiomeric excess, L ligand, Me methyl, cod 1,5-cyclooctadiene, PhMe toluene, DME 1,2-dimethoxyethane, DCM dichloromethane, THF tetrahydrofuran, Et ethyl, BF4 tetrafluoroborate.

butyl group, and consistently good yields and ee were observed (**3s** and **3t**). The tolerance of $sp^2$ C–O bond was also examined and the corresponding products **3u** and **3v** could be furnished in good yields and with good ee, thus providing an opportunity to construct more complex molecules through downstream manipulations[60]. Whereas, lower yield and enantioselectivity were observed when methoxy group was presented at the para-position, likely due to the electronic effect. Moreover, 1-naphthylmethyl Grignard reagent proved to be a suitable coupling partner under standard conditions (**3w**). Multisubstituted nucleophiles were also successfully used in the enantioselective cross-coupling, and offered a promising entry to achieve axially chiral arenols with a large steric hindrance.

Beside arylmethyl Grignard reagents, the enantioselective cross-coupling of substrate **1a** with *tert*-butyl methyl Grignard reagent was conducted and delivered axially chiral compound **3ab**

in 95% yield and with 99% ee. Futhermore, we also investigated the reactivity of alkyl Griganrd reagents bearing $\beta$-H under the standard conditions. For example, the alkyl Grignard reagent, such as ethyl magnesium bromide and cyclo-hexyl magnesium bromide, were examined under the standard conditions. The sequential analysis of reaction mixtures revealed that no alkylative cross-coupling products were observed. However, due to the agostic interaction between the nickel cenetr and $\beta$-H[61], the formal reduction compound **3ac** via competitive $\beta$-H eliminnation were observed[22,62]. Moreover, phenyl Grignard reagent was examinded under standard reaction conditions, the desired product **3ad** was isolated in 88% yield, while only 20% ee was observed.

**Mechanistic investigation.** During the enantioselective alkylative activation of aromatic C–O bond, ligand **L1** was found as a powerful ligand to furnish the target products in high yields and

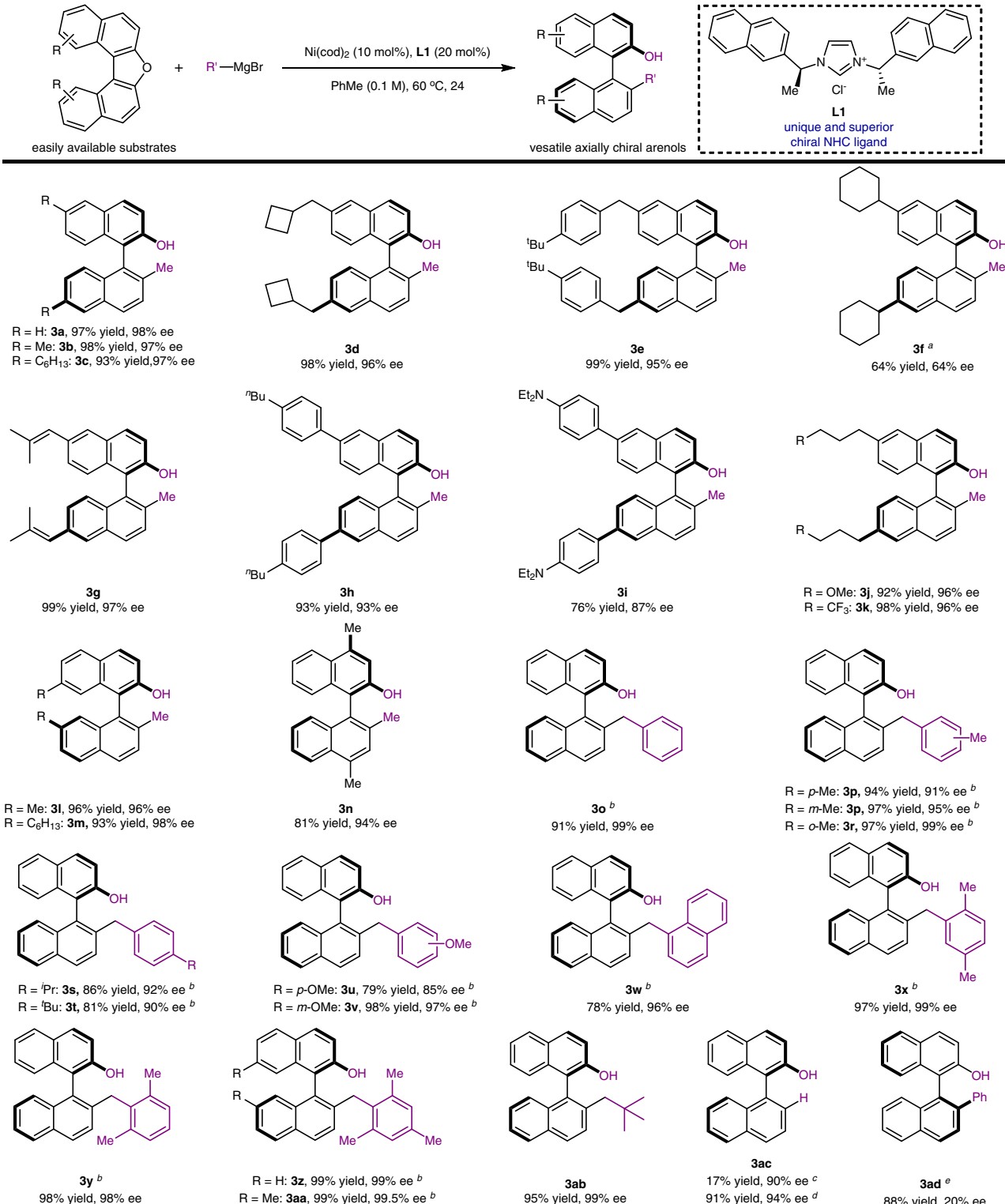

**Fig. 2 Substrate scope of the enantioselective alkylative cross-coupling of unactivated C–O bond.** Unless otherwise stated, the reactions were carried out by using C–O electrophiles (0.2 mmol), R′–MgBr (3.0 equiv.), Ni(cod)₂ (10 mol%), **L1** (20 mol%) in PhMe (0.1 M) at 60 °C for 24 h; [a]The reaction was conducted at 80 °C; [b]R′–MgCl (3.0 equiv.) were used and reactions were conducted at 60 °C for 36 h; [c]Ethyl magnesium bromide (Et-MgBr) (3.0 equiv.) was used; [d]Cyclo-hexyl magnesium bromide (C₆H₁₁-MgBr) (3.0 equiv.) was used; [e]Phenylmagnesium chloride (Ph-MgCl) (3.0 equiv.) was used.

with high ee. To elucidate the role of ligand **L1** in this enantio-selective alkylative cross-coupling, we set out to explore the reaction mechanism. Given the dramatic influence of ligation state of nickel in transition-metal catalyzed aromatic C–O activation[63–66], control experiments were conducted to identify

nickel's ligation state in the enantioselective alkylative cross-coupling of aromatic C–O electrophiles. By using 5 mol% Ni(cod)₂ as the nickel source, the enantioselective methylative activation of substrate **1a** were performed in the presence of different amount of chiral ligand **L1**, and found that only 76%

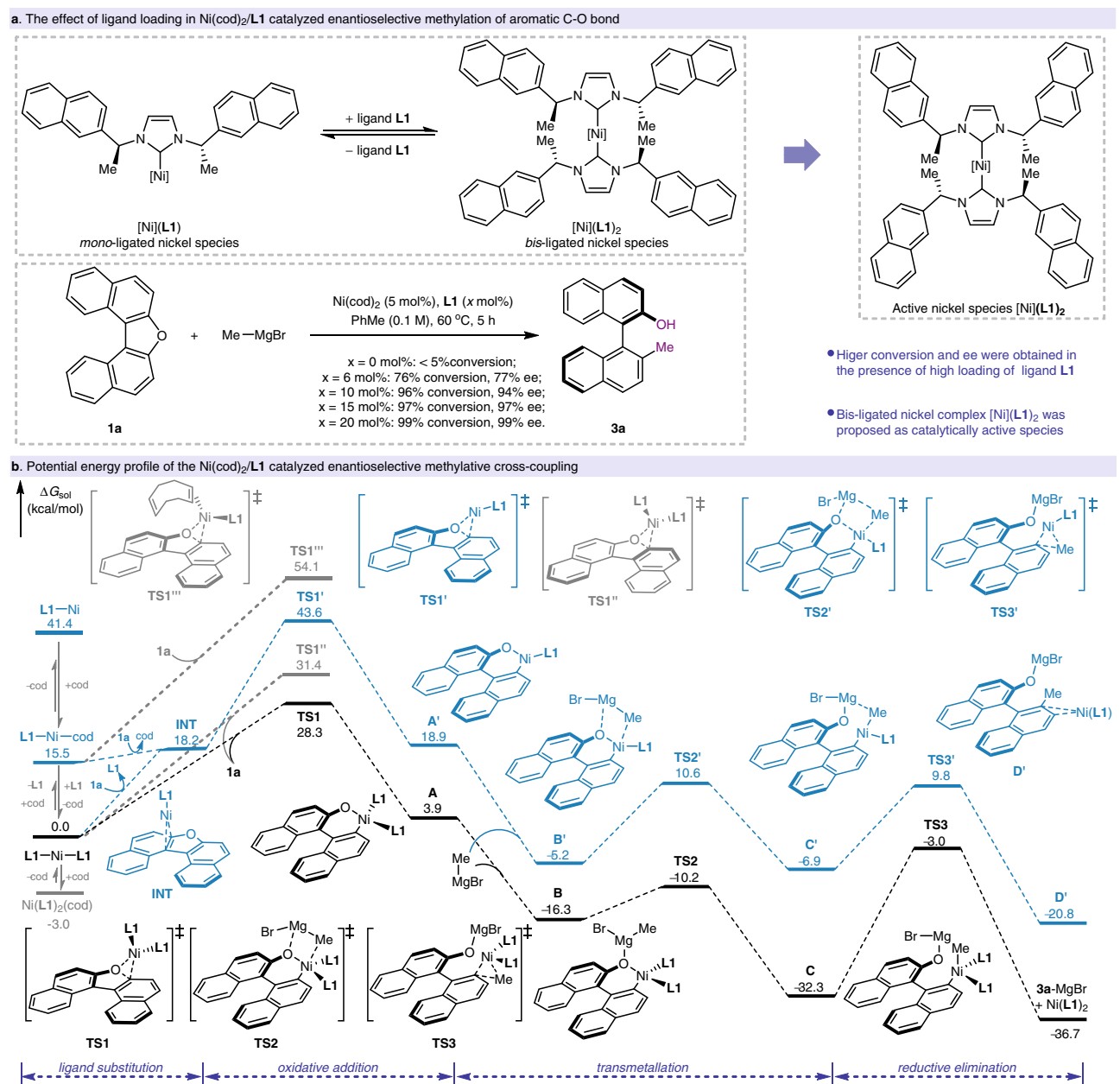

**Fig. 3 Mechanistic investigation. a** The effect of ligand loading in Ni(cod)₂/**L1** catalyzed enantioselective methylation of aromatic C–O bond; **b** Potential energy profile (PES) of the Ni(cod)₂/**L1** catalyzed enantioselective methylative cross-coupling. Black: PES of the Ni(**L1**)₂ catalyzed enantioselective methylative cross-coupling; Blue: PES of the Ni(**L1**) catalyzed enantioselective methylative cross-coupling.

conversion and 77% ee were observed when 6 mol% of ligand **L1** was used as the ligand, in contrast, 99% conversion and 99% ee could be obtained by using 20 mol% ligand **L1** (Fig. 3a), thus suggesting that bis-ligated nickel complex might serve as the catalytically active species in the nickel-catalyzed enantioselective alkylative cross-coupling. In contrast, mono-ligated nickel complex were proposed as the catalytically active species in our previously reported enantioselective arylative activation of aromatic C–O bond (see Supplementary Table 3)[54,67].

Furthermore, conputational studies of nickel-catalyzed enantioselective methylation of substrate **1a** were performed to understand the detailed mechanism (Fig. 3b and Supplementary Data 1). Our calculation results revealed that the bis-ligated nickel complex Ni(**L1**)₂ is significantly more stable than the mono-**L1**-ligated species Ni(**L1**)(cod) and Ni(**L1**) by 15.5 and 41.4 kcal/mol, respectively. Alternatively, the mono-**L1**-ligated species **INT**

could be formed via ligand substitution between substrate **1a** and complex Ni(**L1**)₂ and this process was obviously endergonic by 18.2 kcal/mol. Although Ni(**L1**)₂(cod) is slightly more stable than Ni(**L1**)₂ by 3.0 kcal/mol, the cod ligand is not essential for the enantioselective cross-coupling due to high yield and high enantioselectivity could be obtained by using NiBr₂(dme) as the nickel source (Table 1, entry 4). Therefore, computaional studies on mechanism through bis-ligated pathway were initially conducted. From the nickel complex Ni(**L1**)₂, an oxidative addition of C–O bond to Ni(**L1**)₂ could take place via **TS1** with an energy barrier of 28.3 kcal/mol and generated a six-membered metallo-cycle **A**. Futher coordination of oxygen to magnesium is exergonic by 20.2 kcal/mol as compared to the intermediate **A**. Thereafter, transmetalation easily occurs via **TS2** with an energy barrier of 6.1 kcal/mol, leading to the intermediate **C**. The C(sp²)–C(sp³) reductive elimination via **TS3** eventually affords

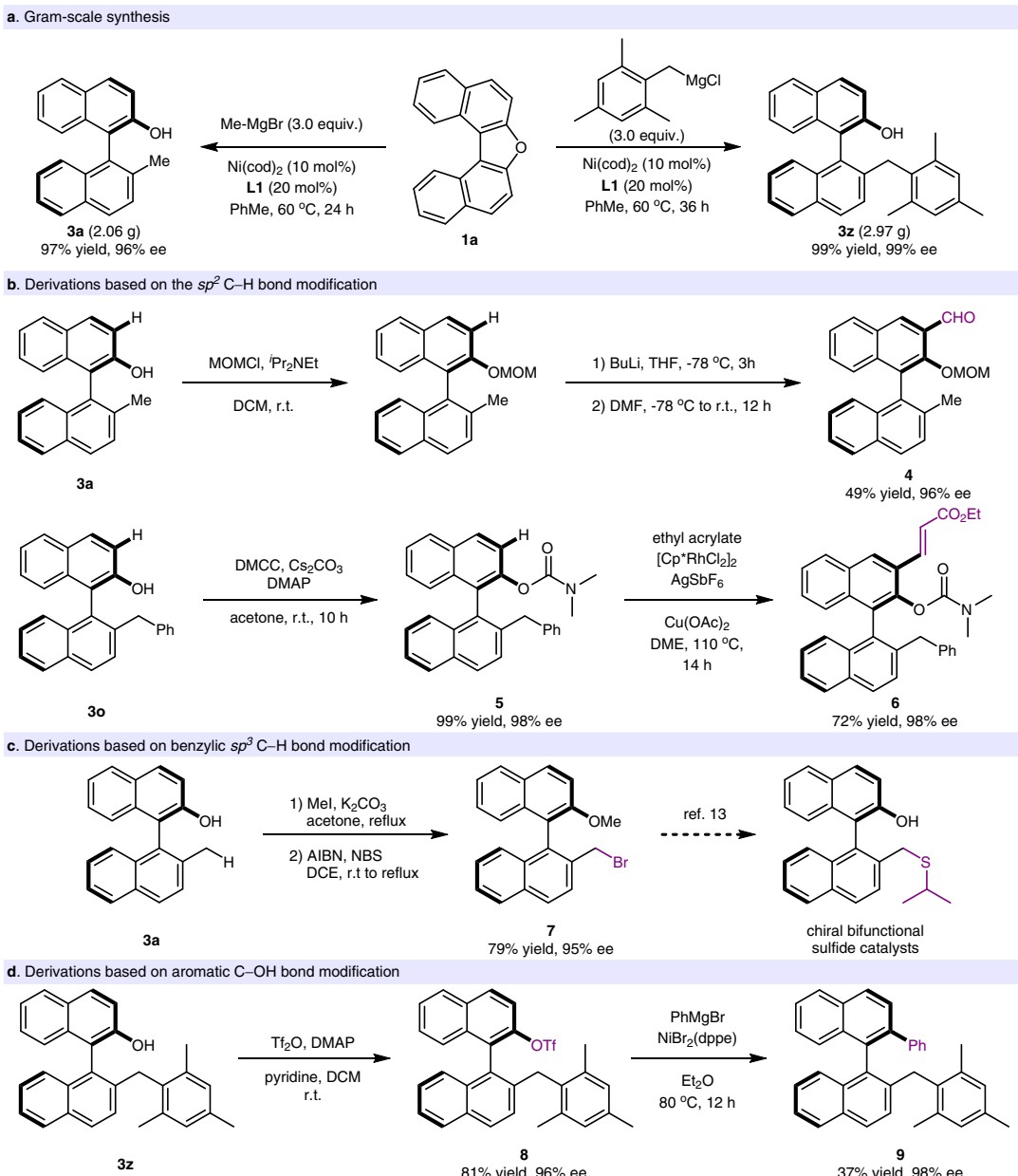

**Fig. 4 Gram-scale synthesis and derivation experiments. a** Gram scale synthesis. **b** Derivations based on the *sp²* C–H bond modification. **c** Derivations based on benzylic *sp³* C–H bond modification. **d** Derivations based on aromatic C–OH bond modification. MOMCl chloromethyl methyl ether, DMF dimethylformamide, DMCC dimethylcarbamyl chloride, DMAP 4-dimethylaminopyridine, AIBN azobisisobutyronitrile, NBS N-bromosuccinimide, dppe 1,2-bis(diphenylphosphino)ethane.

the product **3a**-MgBr and regenerates the bis-ligated nickel species Ni(**L1**)₂ for the next catalytic cycle. Notably, the stereoisomer transition state **TS1″** was calculated to have a higher free energy than **TS1** by 3.1 kcal/mol, which is in accordance with the stereochemical outcome of the enantioselective alkylative cross-coupling. For comparison, DFT calculations on the catalytic cycle via mono-ligated pathway were also carried out. The results showed that the oxidative addition of C–O bond to Ni(**L1**)(cod) via **TS1** undergoes very high energy barrier ($\Delta G^{\ddagger} = 54.1$ kcal/mol) and thus the cod-free pathway was considered for mono-ligated process. As shown in Fig. 3b (blue line), the mono-ligated pathway undergoes similar steps as in the case of bis-ligated one. However, the whole potential energy surface for mono-ligated pathway is located above that of the bis-ligated one, which is in line with our control experiments that high loading of ligand **L1** was required for good catalytic performance and further indicates

that the enantioselective alkylative cross-coupling was facilitated via a bis-ligated pathway. In the case of ligand **L2**, a similar bis-ligated pathway was located (see Supplementary Fig. 22), and found that the initial oxidative addition of C–O bond to Ni(**L2**)₂ complex needs to overcome a high energy barrier of 38.5 kcal/mol due to the steric hindrance, which probably accounts for the poor performance of nickel-catalyzed enantioselective methylation of aromatic C–O bond by using ligand **L2**.

Gram-scale synthesis and derivation experiments. The enantioselective C(sp³)–C(sp²) cross-coupling protocol could be easily scaled up to gram-scale without a detrimental effect (Fig. 4a). For example, 2.06-gram of compound **3a** could be obtained in 97% yield with 96% ee. Moreover, the axially chiral arenol **3z**, which holds large steric hindrance, could be prepared on 2.97-gram scale with 99% ee. To demonstrate the application potential of this enantioselective cross-coupling, several enantioretentive

**Fig. 5 Rational expansion for the synthesis of deuterated axially chiral compounds of type A.** The reactions were carried out by using C–O electrophiles (1.0 equiv.), CD$_3$–MgI (4.0 equiv.), Ni(cod)$_2$ (10 mol%), **L1** (20 mol%), MgBr$_2$ (5.0 equiv.) in PhMe (0.1 M) at 80 °C for 24 h. Isolated yields were reported, ee was determined by using chiral HPLC.

transformations were conducted based on the three modifiable sites (Fig. 4b). For example, by using the $sp^2$ C–H modifiable site, the aldehyde **4**, which presents as a precursor to prepare salen and salan type ligands, could be furnished in 49% yield without any erosion of enantioselectivity via an intermediate of methoxymethyl masked **3a**. Moreover, compound **3o** was treated with dimethyl carbamoyl chloride (DMCC) and the generated phenol carbamate **5** was submitted into a rhodium-catalyzed olefinative C($sp^2$)-H activation[68], and afforded axially chiral compound **6** in 72% yield and with 98% ee.

With N-bromosuccinimide (NBS) as the bromination reagent, azobisisobutyronitrile (AIBN) mediated bromination of benzylic C–H position could occur selectively, and delivered the corresponding axially chiral benzylic bromide **7** in 79% yield and with comparable enantioselectivity (Fig. 4c). The benzylic bromide has been viewed as a versatile handle for further transformations, such as the preparation of chiral aminophenol catalysts and bifunctional sulfide catalysts[31,32].

On the basis of C–OH modifiable site, triflation experiment of **3z** with trifluoromethanesulfonic anhydride (Tf$_2$O) was performed and gave **8** in 81% yield with 96% ee. Further derivations based on OTf group were conducted. For example, arylative cross-coupling of compound **8** with phenyl Grignard reagent gave compound **9** in 37% yield and with 98% ee (Fig. 4d). Of particular note, starting from axially chiral arenols, atropisomeric aniline could be directly furnished using 2-bromopropanamide as the amination reagent according to Guo's method[69], further transformation of the atropisomeric aniline could offer axially chiral biaryl iodide[44]. These derivations and many others clearly certified that the developed enantioselective cross-coupling via aromatic C–O bond cleavage can serve as a useful platform to access diverse axially chiral skeletons.

Rational expansion. Inspired by the increasing interest of deuterated "magic methyl" group in pharmaceutical development[70], CD$_3$–MgI was prepared by using easily available deuterated iodomethane and was subjected into the enantioselective catalytic cross-coupling of electrophile **1a**, however, only 21% conversion was observed under standard reaction conditions. We envisioned that the addition of MgBr$_2$ might improve the efficiency of this transformation by generating CD$_3$–MgBr in-situ and facilitate the cleavage of aromatic C–O bond as an Lewis acid[23]. Thus, a modified reaction conditions by adding MgBr$_2$ (5.0 equiv.) and raising the reaction temperature to 80 °C was established, and facilitated the construction of compound **3ae** in 55% yield with 89% ee (Fig. 5). Other electrophiles were also examined and the corresponding products were observed with satisfied results (**3af** and **3ag**).

In summary, we have developed an efficient C($sp^3$)–C($sp^2$) cross-coupling protocol via enantioselective cleavage of unactivated aromatic C–O bond. By using a unique chiral N-heterocyclic carbene (NHC) ligand, this cross-coupling is facilitated efficiently via nickel catalysis with good enantiocontrol (up to 99% yield and 99.5% ee). The mechanistic investigation suggests that bis-ligated nickel complex might be served as catalytically active species in this transformation. Moreover, CD$_3$ group could also be successfully introduced into the targeted molecules under a simply modified reaction conditions. This chemistry features easy availability of starting materials and simple conditions. Of particular note, further derivations of products based on $sp^2$ C–H, $sp^3$ C–H, and C–OH modifiable sites demonstrate the huge application potential of the developed methodology as a useful platform to achieve diverse axially chiral molecules, thus its widespread application in complex molecule synthesis and asymmetric catalysis could be anticipated.

## Methods

Representative procedure for the synthesis of axially chiral compound **3a**. In a nitrogen-filled glovebox, Ni(cod)$_2$ (5.6 mg, 10 mol%), **L1** (8.3 mg, 20 mol%) were added to an oven-dried 15 mL sealed-tube which was charged with a stir bar. Next, the Grignard reagent **2a** in 2-Me-THF (3.0 M, 0.2 mL, 3.0 equiv.) was added via syringe. The vial was capped and the mixture was stirred at room temperature for 10 min, at which time it was a brown homogeneous solution. The solvent 2-Me-THF was removed under vacuum. Then, the C–O electrophile **1a** (53.7 mg, 0.2 mmol) and anhydrous toluene (0.1 M) was added to the vial, and the cap was sealed. The mixture was stirred at 60 °C for 24 h. The reaction mixture was quenched with aqueous HCl (1.0 M), and washed with EtOAc (3.0 mL * 2). The organic layers were combined, the mixture was concentrated, and the residue was purified by flash chromatography on silica gel. The reaction afforded compound **3a** with 97% isolated yield as a white solid. All new compounds were fully characterized (See Supplementary Method).

## Data availability

The data that support the findings of this study are available within the article and its Supplementary Information files. Correspondence and requests for materials should be addressed to G.L. (luogen@ahu.edu.cn) and Z.C.C. (zc_cao@ahau.edu.cn).

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

## Acknowledgements

We are grateful for financial support from the Anhui Agricultural University, Anhui Provincial Natural Science Foundation (Grant Nos. 2108085QC119 to Z.C.C. and 2108085Y04 to G. L.) and National Natural Science Foundation of China (Grant No. 22003001 to G. L.).

## Author contributions

Z.Z., J.Z. and Q.G. contributed equally to this work. Z.Z., J.Z. and Q.G. performed the experiments and analyzed the data. M.Y., H.C. and T.S. assisted the purification of compounds and analysis of data. Y. Z. and G.L. performed DFT calculations. Z.C.C. designed and directed the whole project and wrote the manuscript.

## Competing interests

The authors declare no competing interests.
