## [Peer Review File · Nature Communications]

REVIEWER COMMENTS

Reviewer #1 (Remarks to the Author):

The manuscript by Cao and co-workers reports the enantioselective C(sp³)-C(sp²) cross-coupling of unactivated aromatic C-O electrophiles. The reported method, inspired by their previous Nickel-catalyzed enantioselective arylation of aromatic C-O bond (J. Am. Chem. Soc. 2021, 143, 18380.), enables the direct synthesis of axially chiral 2-alkyl-2'-hydroxyl-biaryl compounds with good enantioselectivity. Despite this protocol provides advances in asymmetric alkylative activation of aromatic C-O bond, the manuscript presents critical points that need to be further clarified.

(1) At the development of their methodology, the authors identified L1 as a unique ligand that gives the best yield and high enantioselectivity. By contrast, L2, a similar NHC ligand used in the arylation of aromatic C-O bond, leads to poor yield (<10%). Why do alkylation and arylation reactions prefer different ligands? What is the unique role of L1 on the reactivity and enantioselectivity of alkylation reaction?

(2) The author claimed in the introduction that the challenge for enantioselective C(sp³)-C(sp²) cross-coupling of unactivated aromatic C-O electrophiles may be due to the unfavorable reductive elimination of high-valent transition-metal complex. Does the alkylative reaction involve any high-valence Nickel complex? Combined with question 1, the authors should provide more mechanistic insight of reaction via mechanistic experiments or computational study.

(3) The nucleophile used in the reported method seems limited to methyl, arylmethyl, and tertbutyl methyl Grignard reagents, which only contain α hydrogen atoms. Is the approach compatible with the presence of β hydrogen atoms? i.e. Is the method applicable to ethyl, propyl, and other alkyl Grignard reagents? If not, do the authors have any explanations?

Other minor modifications required:

1. The substituent's labels in Fig 2 are misleading. The substituents of dinaphthylfuran and Grignard reagents share the same label R, but they are not always the same. The two substituents keep the same in all dinaphthylfuran derivatives and do not need two different labels.
2. Some of the journals listed in the references are not appropriately abbreviated. For instance, ref 24, 31, and 60.

Reviewer #2 (Remarks to the Author):

This manuscript by Cao has illustrated the atroposelective synthesis of 2-alkyl-2'-hydroxyl-biaryls by C-O bond activation of diarylfurans via nickel catalysis. This work is the extension of their recent publication about the enantioselective arylation of aromatic C-O bond (J. Am. Chem. Soc. 2021, 143, 18380). Although these two works looked quite similar, reactions described in this manuscript represented rare cases of the enantioselective C(sp³)-C(sp²) cross-coupling of unactivated aromatic C-O electrophiles. It provided quite convenient access to 2-alkyl-2'-hydroxyl-biaryl scaffolds, which are prevalent in core structures of several types of natural products, chiral ligands and catalysts, as the authors described in Figure 1c and section C of the Supporting Information. Therefore, my feeling is that this manuscript is generally suitable to be published in *Nature Communications*.

However, there are several very important points that the authors should address before its publication.

1. The authors have strongly claimed that a different N-substituted NHC ligand was unique and critical to realize the current alkylation instead of previous arylation. However, they have not explained the unique effect of the ligand at all in the manuscript. This reviewer would suggest the authors to do some experiments or conduct DFT calculations to explain this interesting issue.
2. In Line 30, it is quite strange to describe the oxygen-based compounds are nontoxic. The authors should revise this.
3. In Line 33, "Scheme 1b" should be "Fig. 1b".
4. In the title of Fig. 1d, "route" should be "synthetic route".
5. In Line 83, "skekeleton" should be "skeleton".
6. In Entry 6, Table 1, L2-L8 and L1 should be bold.
7. In Line 113, "was" should be "were".
8. In Line 142, "high ee's" should be "high ee".
9. In Line 143, methyl, isopropyl and tert-butyl should be methyl group, isopropyl group and tert-butyl group.
10. As for the data, 30% of the ¹H NMR and ¹³C NMR spectra in the Supporting Information were not clean; there are very high residue peaks of solvents, grease and impurities in these spectra. Furthermore, 50% of the HPLC spectra were not good enough to determine the high optical purities of the substrates. The authors should revise them to meet the high standards of this top journal.

Reviewer #3 (Remarks to the Author):

Comments to the Author

This manuscript reports the development of enantioselective C(sp³)-C(sp²) cross-coupling reaction via aromatic C-O bond activation of dinaphthylfuran derivatives with alkyl magnesium bromides. All these catalytic transformations were attained by employing 10 mol% Ni(cod)₂ and chiral NHC ligand L (20 mol%) to generate desirable product axially chiral biaryl arenol 3 in good yield with high ee. A wide range of substrates including numerous arylmethyl Grignard bromide reagents were examined and found suitable in this reaction protocols. Authors further demonstrated the utilities of these chiral products of 3 by performing derivation modifications based on sp² C-H bond activation, sp³ C-H bond activation and phenol functionalization. The work is regarded as original significance to the synthetic field based on C-O bond activation, as it has established an enantioselectivity in this method. Overall, this work can be considered for publication in this top journal if these following questions needed to be addressed in satisfactory manner with additional data.

Questions and suggestions:

1. Explain the failure of the secondary alkyl Grignard reagent to have alkylative cross-coupling product, instead obtaining compound 3ac.
2. Long alkyl and aryl Grignard reagents must include for this publication.
3. lack of detailed of mechanism studies and proposal mechanism
4. Computational analysis should be included to support the mechanism studies.
5. In the author's previous work (Fig. 1e), they use L2 ligand for enantioselective arylation cross coupling reaction. However, only L1 works for alkylation in similar reaction condition. The difference between L1 and L2 only at the connection position of alpha and beta carbon of naphthalene to NHCs. What is the reason that L2 is more efficient than L1?

6. What is the reason that the author used 1:2 catalyst/ligand ratio in this reaction instead of 1:1.2 ratio in previous work (Fig. 1e)?

7. Although the author don't explain the mechanism in this research, they proposed direct oxidative addition of Ni(0)/L to C-O bond of dinaphthylfuran in previous report (Fig. 1e). However, they modified the reaction condition by adding MgBr₂ in rational expansion on deuterated reaction, which gave a possible new mechanism for Lewis acid activated C-O activation (the mechanism in Chin. J. Chem. 2018,36,183).

8. How about dibenzofuran?

minor issue:

1. Type error in the title of Fig. 1d: rout"e".
2. The bond type of C-O should be consisted in all manuscript.
3. CD₃-MgBr instead of MeMgBr in the footnote of Fig. 4.
4. Wrong compound label for compound 3s in Page S40 of supporting information.

REVIEWER COMMENTS

Reviewer #1 (Remarks to the Author):

The manuscript by Cao and co-workers reports the enantioselective C(sp³)-C(sp²) cross-coupling of unactivated aromatic C-O electrophiles. The reported method, inspired by their previous Nickel-catalyzed enantioselective arylation activation of aromatic C-O bond (*J. Am. Chem. Soc.* **2021**, *143*, 18380.), enables the direct synthesis of axially chiral 2-alkyl-2'-hydroxyl-biaryl compounds with good enantioselectivity. *Despite this protocol provides advances in asymmetric alkylative activation of aromatic C-O bond, the manuscript presents critical points that need to be further clarified.*

(1) At the development of their methodology, the authors identified **L1** as a unique ligand that gives the best yield and high enantioselectivity. By contrast, **L2**, a similar NHC ligand used in the arylation activation of aromatic C-O bond, leads to poor yield (<10%). Why do alkylation and arylation reactions prefer different ligands? What is the unique role of **L1** on the reactivity and enantioselectivity of alkylation reaction?

(2) The author claimed in the introduction that the challenge for enantioselective C(sp³)-C(sp²) cross-coupling of unactivated aromatic C-O electrophiles may be due to the unfavorable reductive elimination of high-valent transition-metal complex. Does the alkylative reaction involve any high-valence Nickel complex? Combined with question 1, the authors should provide more mechanistic insight of reaction via mechanistic experiments or computational study.

(3) The nucleophile used in the reported method seems limited to methyl, arylmethyl, and tertbutyl methyl Grignard reagents, which only contain α hydrogen atoms. Is the approach compatible with the presence of β hydrogen atoms? i.e. Is the method applicable to ethyl, propyl, and other alkyl Grignard reagents? If not, do the authors have any explanations?

Other minor modifications required:

1. The substituent's labels in *Fig 2* are misleading. The substituents of dinaphthylfuran and Grignard reagents share the same label R, but they are not always the same. The two substituents keep the same in all dinaphthylfuran derivatives and do not need two different labels.
2. Some of the journals listed in the references are not appropriately abbreviated. For instance, ref 24, 31, and 60.

Reviewer #2 (Remarks to the Author):

This manuscript by Cao has illustrated the atroposelective synthesis of 2-alkyl-2'-hydroxyl-biaryls by C-O bond activation of diarylfurans via nickel catalysis. This work is the extension of their recent publication about the enantioselective arylation activation of aromatic C-O bond (*J. Am. Chem. Soc.* **2021**, *143*, 18380). Although these two works looked

quite similar, reactions described in this manuscript represented rare cases of the enantioselective C(sp³)-C(sp²) cross-coupling of unactivated aromatic C-O electrophiles. It provided quite convenient access to 2-alkyl-2'-hydroxyl-biaryl scaffolds, which are prevalent in core structures of several types of natural products, chiral ligands and catalysts, as the authors described in *Figure 1c* and section C of the Supporting Information. *Therefore, my feeling is that this manuscript is generally suitable to be published in nat commun.*

However, there are several very important points that the authors should address before its publication.

1. The authors have strongly claimed that a different N-substituted NHC ligand was unique and critical to realize the current alkylation instead of previous arylation. However, they have not explained the unique effect of the ligand at all in the manuscript. This reviewer would suggest the authors to do some experiments or conduct DFT calculations to explain this interesting issue.
2. In Line 30, it is quite strange to describe the oxygen-based compounds are nontoxicity. The authors should revised this.
3. In Line 33, "Scheme 1b" should be "Fig. 1b".
4. In the title of *Fig. 1d*, "rout" should be "synthetic route".
5. In Line 83, "skekeleton" should be "skeleton".
6. In Entry 6, Table 1, **L2-L8** and **L1** should be bold.
7. In Line 113, "was" should be "were".
8. In Line 142, "high ee's" should be "high ee".
9. In Line 143, methyl, isopropyl and tert-butyl should be methyl group, isopropyl group and tert-butyl group.
10. As for the data, 30% of the ¹H NMR and ¹³C NMR spectra in the Supporting Information were not clean; there are very high residue peak of solvents, grease and impurities in these spetra. Furthermore, 50% of the HPLC spectra were not good enough to determine the high optical purities of the substrates. The authors should revise them to meet the high standards of this top journal.

Reviewer #3 (Remarks to the Author):

Comments to the Author

This manuscript reports the development of enantioselective C(sp³)-C(sp²) cross-coupling reaction via aromatic C-O bond activation of dinaphthylfuran derivatives with alkyl magnesium bromides. All these catalytic transformations were attained by employing 10 mol% Ni(cod)₂ and chiral NHC ligand L (20 mol%) to generate desirable product axially chiral biaryl arenol **3** in good yield with high ee. A wide range of substrates including numerous arylmethyl Grignard bromide reagents were examined

and found suitable in this reaction protocols. Authors further demonstrated the utilities of these chiral products of **3** by performing derivation modifications based on sp^2 C-H bond activation, sp^3 C-H bond activation and phenol functionalization. The work is regarded as original significance to the synthetic field based on C-O bond activation, as it has established an enantioselectivity in this method. *Overall, this work can be considered for publication in this top journal if these following questions needed to be addressed in satisfactory manner with additional data.*

Questions and suggestions:

1. Explain the failure of the secondary alkyl Grignard reagent to have alkylative cross-coupling product, instead obtaining compound **3ac**.
2. Long alkyl and aryl Grignard reagents must include for this publication.
3. lack of detailed of mechanism studies and proposal mechanism
4. Computational analysis should be included to support the mechanism studies.
5. In the author's previous work (Fig. 1e), they use **L2** ligand for enantioselective arylation cross coupling reaction. However, only **L1** works for alkylation in similar reaction condition. The difference between **L1** and **L2** only at the connection position of alpha and beta carbon of naphthalene to NHCs. What is the reason that **L1** is more efficient than **L2**?
6. What is the reason that the author used 1:2 catalyst/ligand ratio in this reaction instead of 1:1.2 ratio in previous work (Fig. 1e)?
7. Although the author don't explain the mechanism in this research, they proposed direct oxidative addition of Ni(0)/L to C-O bond of dinaphthylfuran in previous report (Fig. 1e). However, they modified the reaction condition by adding MgBr₂ in rational expansion on deuterated reaction, which gave a possible new mechanism for Lewis acid activated C-O activation (the mechanism in *Chin. J. Chem.* **2018**, 36, 183).
8. How about dibenzofuran?

minor issue:

1. Type error in the title of Fig. 1d: rout"e".
2. The bond type of C-O should be consisted in all manuscript.
3. CD₃-MgBr instead of MeMgBr in the footnote of Fig. 4.
4. Wrong compound label for compound **3s** in Page S40 of supporting information.

Point-by-point response to the reviewers' comments

Reviewer #1 (Remarks to the Author):

The manuscript by Cao and co-workers reports the enantioselective C(sp³)-C(sp²) cross-coupling of unactivated aromatic C-O electrophiles. The reported method, inspired by their previous Nickel-catalyzed enantioselective arylation activation of aromatic C-O bond (*J. Am. Chem. Soc.* 2021, 143, 18380.), enables the direct synthesis of axially chiral 2-alkyl-2'-hydroxyl-biaryl compounds with good enantioselectivity. *Despite this protocol provides advances in asymmetric alkylative activation of aromatic C-O bond, the manuscript presents critical points that need to be further clarified.*

General response: Thanks for the reviewer's positive comments. Your kind suggestions help us a great deal in improving the quality of our manuscript. Please see below for our detailed response.

Comment: At the development of their methodology, the authors identified **L1** as a unique ligand that gives the best yield and high enantioselectivity. By contrast, **L2**, a similar NHC ligand used in the arylation activation of aromatic C-O bond, leads to poor yield (<10%). Why do alkylation and arylation reactions prefer different ligands? What is the unique role of **L1** on the reactivity and enantioselectivity of alkylation reaction?

*Response: Thanks for the reviewer's question. According to the reviewer's comments, we set out to explore the mechanism for the nickel catalyzed alkylative cross-coupling and understand the role of ligand **L1** in the alkylative cross-coupling.*

*The ligation state of nickel in the enantioselective methylative cross-coupling was initially investigated. By using 5 mol% Ni(cod)₂ as the pre-catalyst, we performed the enantioselective methylative cross-coupling of **1a** in the presence of different amount of ligand **L1**, and found that higher conversion and ee were observed when high loading of ligand **L1** was used. For example, the target product was generated in only 76% conversion and with 77% ee when 6 mol% ligand **L1** was used, while 99% conversion and 99% ee were obtained by using 20 mol% ligand **L1** (Figure R1-1). These results indicated that bis-ligated nickel complexes [Ni](**L1**)₂ might serve as the catalytically active species in the enantioselective C(sp³)-C(sp²) cross-coupling.*

*DFT calculations on the enantioselective methylation of aromatic C-O bond of **1a** were performed to understand the mechanism. As shown in Figure R1-2, the bis-ligated nickel species Ni(**L1**)₂ is more stable than the mono-ligated nickel species INT by 18.2 kcal/mol. From the complex Ni(**L1**)₂, oxidative addition of C-O bond to Ni(**L1**)₂ could occur via TS1 with an energy barrier of 28.3 kcal/mol and generated a 6-membered metallocycle A. Subsequently, the coordination of oxygen-atom with magnesium atom was quite exergonic by 20.2 kcal/mol as compared to intermediate A. Subsequent transmetalation occurred easily via TS2 with an energy barrier of 6.1 kcal/mol, leading to the intermediate C. Then, C(sp²)-C(sp³) reductive elimination via TS3 eventually afforded the product **3a-MgBr** and regenerated the bis-ligated nickel species Ni(**L1**)₂ for the next catalytic cycle. To directly compare with the bis-ligated mechanism, the DFT*

calculations on catalytic cycle through mono-ligated pathway were also carried out (Figure R1-2, blue line). However, the whole potential energy surface for mono-ligated one is located above that of the bis-ligated one. Therefore, the catalytic cycle maybe take place through bis-ligated pathway. This is agree with the control experiments that high loading of ligand L1 is required for good catalytic performance.

Figure R1-1. The effect of loading of ligand L1

Figure R1-2. PES for the overall catalytic cycle for Ni/L1 catalyzed enantioselective methylation

Moreover, to understand the unsuccessful methylative cross-coupling of substrate **1a** by using L2 as the ligand, DFT calculations were performed (Figure R1-3), and found that the initial oxidative addition of C–O bond to Ni(L2)₂ needs to overcome a high energy barrier of 38.5 kcal/mol due to steric hindrance, which probably account for the poor performance of Ni(cod)₂/L2 catalyzed methylation of aromatic C–O bond.

Figure R1-3. PES for the overall catalytic cycle for Ni/L2 catalyzed enantioselective methylation

By comparison, we also examined the nickel's ligation state in enantioselective arylation of aromatic C-O bond (Figure R1-4). By using 10 mol% Ni(cod)₂ as the precatalyst, the arylation cross-coupling of C-O electrophile **1a** were conducted in the presence of different amount of ligand **L2**, and found that lower conversion of substrate **1a** was observed when high loading of **L2** was used, these results suggested that *mono-ligated nickel species maybe serve as the catalytically active species in the arylation cross-coupling*, which was consistent with the proposed mechanism in our previous work (J. Am. Chem. Soc. 2021, 143, 18380). Moreover, the groups of Chatani and Mori found that *energy barrier for transmetalation of phenyl nucleophile with bis-ligated nickel species Ni(L)₂(Ar)(OMe) was much higher than that for mono-ligated nickel species Ni(L)(Ar)(OMe), and one ligand need to occur dissociation in the transition state* (J. Am. Chem. Soc. 2017, 139, 30, 10347–10358), while high concentration of ligand will be unfavorable for the ligand dissociation during transmetalation step in cross-coupling. Thus, the different nickel's ligation states and the steric hindrance of ligand made arylation and alkylation preferring ligand **L1** or ligand **L2**, and **L1** could facilitate the alkylation cross-coupling as a unique ligand through a bis-ligated pathway.

Figure R1-4. The ligation state of nickel in arylation cross-coupling

Comment: The author claimed in the introduction that the challenge for enantioselective C(sp³)-C(sp²) cross-coupling of unactivated aromatic C-O electrophiles may be due to the unfavorable reductive elimination of high-valent transition-metal complex. Does the alkylation reaction involve any high-valence Nickel complex? Combined with question 1,

the authors should provide more mechanistic insight of reaction via mechanistic experiments or computational study.

Response: Thanks for the reviewer's constructive comments. According to the reviewer's comments, DFT calculations on mechanism were conducted (Figure R1-2), and revealed that the nickel catalyzed enantioselective alkylative cross-coupling was accomplished via Ni(0)/Ni(II) catalytic cycle. Bis-ligated nickel complexes, such as intermediate $(L1)_2Ni^{(II)}(Ar)(OR)$ (intermediate **A** in Figure R1-2) and $(L1)_2Ni^{(II)}(Ar)(Me)$ (intermediate **B** in Figure R1-2) might be formed as the catalytically active species via oxidative addition and transmetalation steps, and no high-valent Ni(III) species and Ni(IV) species were formed. On the basis of these results, we have re-written this sentence by deleting the words "high valent" in the revised manuscript, and more references were cited to support the revised sentence.

Comment: The nucleophile used in the reported method seems limited to methyl, arylmethyl, and tertbutyl methyl Grignard reagents, which only contain α hydrogen atoms. Is the approach compatible with the presence of β -hydrogen atoms? i.e. Is the method applicable to ethyl, propyl, and other alkyl Grignard reagents? If not, do the authors have any explanations?

Response: Thanks for the reviewer's comments. According to the reviewer's comments, we conducted the nickel catalyzed enantioselective alkylative cross-coupling of substrate **1a** by using different types of alkyl Grignard reagents, including primary alkyl magnesium bromides and secondary alkyl magnesium bromides, while no desired alkylative cross-coupling products were observed (Figure R1-5). Careful analysis of reaction mixtures revealed that the formal reductive was obtained, likely as a result of a competitive β -H elimination was occurred after transmetalation step (see β -H elimination in C-O bond functionalization: *Angew. Chem. Int. Ed.* 2016, 55, 6093-6098; *Chem. Sci.* 2015, 6, 3410-3414; *Chem. Soc. Rev.* 2014, 43, 8081-8097).

Figure R1-5. The enantioselective cross-coupling of electrophile **1a** with nucleophiles with β -H

Comment: The substituent's labels in Fig 2 are misleading. The substituents of dinaphthylfuran and Grignard reagents share the same label R, but they are not always the same. The two substituents keep the same in all dinaphthylfuran derivatives and do not need two different labels.

Response: Thanks for the reviewer's comments, we have corrected the labels in Fig. 2 in the revised manuscript.

Comment: Some of the journals listed in the references are not appropriately abbreviated. For instance, ref 24, 31, and 60.

Response: Thanks for the reviewer's comments, we have corrected these errors in the revised manuscript.

Reviewer #2 (Remarks to the Author):

This manuscript by Cao has illustrated the atroposelective synthesis of 2-alkyl-2'-hydroxyl-biaryls by C-O bond activation of diarylfurans via nickel catalysis. This work is the extension of their recent publication about the enantioselective arylation of aromatic C-O bond (*J. Am. Chem. Soc.* **2021**, *143*, 18380). Although these two works looked quite similar, reactions described in this manuscript represented rare cases of the enantioselective C(sp³)-C(sp²) cross-coupling of unactivated aromatic C-O electrophiles. It provided quite convenient access to 2-alkyl-2'-hydroxyl-biaryl scaffolds, which are prevalent in core structures of several types of natural products, chiral ligands and catalysts, as the authors described in Figure 1c and section C of the Supporting Information. Therefore, my feeling is that this manuscript is generally suitable to be published in *nat commun*.

General response: Thank you for supporting the publication of our manuscript in *Nature Communications*. We have complementally addressed your insightful comments. Please see below:

However, there are several very important points that the authors should address before its publication.

Comment: The authors have strongly claimed that a different N-substituted NHC ligand was unique and critical to realize the current alkylation instead of previous arylation. However, they have not explained the unique effect of the ligand at all in the manuscript. This reviewer would suggest the authors to do some experiments or conduct DFT calculations to explain this interesting issue.

Response: Thanks for the reviewer's comments. According to the reviewer's comments, we conducted mechanistic studies to elucidate the unique role of ligand L1 in nickel catalyzed enantioselective cross-coupling of aromatic C-O electrophiles.

Firstly, control experiments were conducted to identify the nickel's ligation state in the enantioselective methylative cross-coupling, and found that higher conversion and ee were observed when high loading of ligand **L1** was used (Figure R2-1). These results indicated that *bis*-ligated nickel complexes $\text{Ni}(\text{L1})_2$ might serve as the catalytically active species in our enantioselective $\text{C}(\text{sp}^3)\text{-C}(\text{sp}^2)$ cross-coupling of aromatic C-O electrophile.

Figure R2-1. The effect of loading of ligand **L1**

Figure R2-2. PES for the overall catalytic cycle for $\text{Ni}/\text{L1}$ catalyzed enantioselective methylation

Secondly, DFT calculations were performed to understand the mechanism. As shown in Figure R2-2, the *bis*-ligated nickel species $\text{Ni}(\text{L1})_2$ is more stable than the *mono*-ligated nickel species **INT** by 18.2 kcal/mol. From the *bis*-ligated complex $\text{Ni}(\text{L1})_2$, oxidative addition of C-O bond to $\text{Ni}(\text{L1})_2$ could occur via **TS1** with an energy barrier of 28.3 kcal/mol and generated a 6-membered metallocycle **A**. The coordination of oxygen atom with magnesium atom is quite exergonic by 20.2 kcal/mol as compared to the intermediate **A**. Subsequently, transmetalation occurs easily via **TS2** with an energy barrier of 6.1 kcal/mol, leading to the intermediate **C**. Then, the subsequent $\text{C}(\text{sp}^3)\text{-C}(\text{sp}^2)$

$C(sp^3)$ reductive elimination via **TS3** eventually affords the product **3a-MgBr** and regenerates the bis-ligated nickel species $Ni(L1)_2$ for the next catalytic cycle. To directly compare with the bis-ligated mechanism, computational analysis of the catalytic cycle through mono-ligated pathway were also carried out, and the results suggested that the reaction underwent similar steps as in the case of bis-ligated one (Figure R2-2, blue line). However, *the whole potential energy surface for mono-ligated one is located above that of the bis-ligated one*, and transmetalation occurred via **TS2'** with an energy barrier of 15.8 kcal/mol. Therefore, the $(L1)_2$ -ligated nickel complexes maybe serve as the catalytically active species and the catalytic cycle took place through bis-ligated pathway. This is agree with the control experiments that high loading of ligand **L1** is required for good catalytic performance.

By comparison, to understand the unsuccessful methylative cross-coupling of substrate **1a** by using **L2** as the ligand, DFT calculations were performed (Figure R2-3), and found that *the initial oxidative addition of C–O bond to $Ni(L2)_2$ needs to overcome a high energy barrier of 38.5 kcal/mol due to the steric hindrance*, which probably account for the poor performance of $Ni(cod)_2/L2$ catalyzed methylation of aromatic C–O bond. Thus, ligand **L1** could facilitate the alkylative cross-coupling with better results than ligand **L2**.

Figure R2-3. PES for the overall catalytic cycle for $Ni/L2$ catalyzed enantioselective methylation

Comment: In Line 30, it is quite strange to describe the oxygen-based compounds are nontoxicity. The authors should revise this.

Response: Thanks for the reviewer's comments, we have deleted the word "nontoxicity" in the revised manuscript.

Comment: In Line 33, "Scheme 1b" should be "Fig. 1b".

Response: Thanks for the reviewer's comments, we have corrected the typo error in the revised manuscript.

Comment: In the title of Fig. 1d, "rout" should be "synthetic route".

Response: Thanks for the reviewer's comments, we have corrected the typo error in the revised manuscript.

Comment: In Line 83, "skekeleton" should be "skeleton".

Response: Thanks for the reviewer's comments, we have corrected the typo error in the revised manuscript.

Comment: In Entry 6, Table 1, **L2-L8** and **L1** should be bold.

Response: Thanks for the reviewer's comments, we have corrected it in the revised manuscript.

Comment: In Line 113, "was" should be "were".

Response: Thanks for the reviewer's comments, we have corrected it in the revised manuscript.

Comment: In Line 142, "high ee's" should be "high ee".

Response: Thanks for the reviewer's comments, we have corrected it in the revised manuscript.

Comment: In Line 143, methyl, isopropyl and tert-butyl should be methyl group, isopropyl group and tert-butyl group.

Response: Thanks for the reviewer's comments, we have corrected it in the revised manuscript.

Comment: As for the data, 30% of the ¹H NMR and ¹³C NMR spectra in the Supporting Information were not clean; there are very high residue peak of solvents, grease and impurities in these spectra. Furthermore, 50% of the HPLC spectra were not good enough to determine the high optical purities of the substrates. The authors should revise them to meet the high standards of this top journal.

*Response: Thanks for the reviewer's comments. According to the reviewer's comments, we conducted extensive works to purify the products, such as **3c, 3d, 3f, 3h, 3i, 3j, 3k, 3l, 3n, 3o, 3p, 3y, 3z, 3aa, 3ad, 3ae and 3af**, by using distilled petroleum ether. However, at current stage, part of them still contain impurities aroused by the used petroleum ether.*

Reviewer #3 (Remarks to the Author):

This manuscript reports the development of enantioselective C(sp³)-C(sp²) cross-coupling reaction via aromatic C-O bond activation of dinaphthylfuran derivatives with alkyl magnesium bromides. All these catalytic transformations were attained by employing 10 mol% Ni(cod)₂ and chiral NHC ligand L (20 mol%) to generate desirable product axially chiral biaryl arenol **3** in good yield with high ee. A wide range of substrates including numerous arylmethyl Grignard bromide reagents were examined

and found suitable in this reaction protocols. Authors further demonstrated the utilities of these chiral products of **3** by performing derivation modifications based on sp^2 C-H bond activation, sp^3 C-H bond activation and phenol functionalization. The work is regarded as original significance to the synthetic field based on C-O bond activation, as it has established an enantioselectivity in this method. *Overall, this work can be considered for publication in this top journal if these following questions needed to be addressed in satisfactory manner with additional data.*

General response: Thank you for supporting the publication of this manuscript in Nature Communications. We have conducted extensive efforts to address your insightful comments. Please see below:

Comment: Explain the failure of the secondary alkyl Grignard reagent to have alkylative cross-coupling product, instead obtaining compound **3ac**.

Response: Thanks for the reviewer's comments. The failure of the secondary alkyl Grignard reagent to have alkylative cross-coupling product was observed likely as a result of a competitive β -H elimination occurred after transmetalation step, which has been documented in nickel catalyzed C-O bond activation (see β -H elimination in C-O bond functionalization: Angew. Chem. Int. Ed. 2016, 55, 6093; Chem. Sci. 2015, 6, 3410; Chem. Soc. Rev. 2014, 43, 8081).

Comment: Long alkyl and aryl Grignard reagents must include for this publication.

*Response: Thanks for the reviewer's constructive comments. According to the reviewer's comments, the phenyl magnesium chloride was used as a nucleophile by using Ni(cod)₂/L1 catalytic system, the target product was obtained in 88% yield but only with 20% ee (Figure R3-1). Moreover, we also conducted the enantioselective cross-coupling of substrate **1a** with ethyl magnesium bromide and (cyclopentylmethyl)magnesium bromide, while no alkylative cross-coupling products were observed, and the formal reduction product via β -H elimination was obtained. More optimization experiments, such as temperature, solvent, were conducted but failed. These results have been added into the revised manuscript.*

Figure R3-1. Examination of the reactivity of aryl and long alkyl Grignard reagent

Comment: lack of detailed of mechanism studies and proposal mechanism

Response: Thanks for the reviewer's constructive comments. According to your comment, we conducted the mechanistic investigation as below:

Control experiments were initially conducted to elucidate nickel's ligation state, and revealed that higher conversion and ee were observed when high loading of ligand **L1** was used (Figure R3-2). Thus indicating that **bis-ligated nickel complexes** $[\text{Ni}](\text{L1})_2$ might be formed as the **catalytically active species** in the enantioselective $\text{C}(\text{sp}^3)\text{-C}(\text{sp}^2)$ cross-coupling of aromatic C-O electrophile.

Figure R3-2. Proposed mechanism and related mechanistic investigations

Furthermore, DFT calculations were performed (Figure R3-2), and revealed that the bis-ligated nickel species $\text{Ni}(\text{L1})_2$ was more stable than the mono-ligated species $\text{Ni}(\text{L1})$ by 41.4 kcal/mol. Therefore, computational studies on mechanism through bis-ligated pathway were preliminarily conducted. From the nickel complex $\text{Ni}(\text{L1})_2$, oxidative addition of C-O bond to $\text{Ni}(\text{L1})_2$ could take place via **TS1** with an energy barrier of 28.3 kcal/mol and generated a six-membered metallocycle

A. The coordination of oxygen-atom with magnesium atom was quite exergonic by 20.2 kcal/mol as compared to the intermediate A. Then, the transmetalation easily occurred via **TS2** with an energy barrier of 6.1 kcal/mol, leading to the intermediate C. The C(sp²)-C(sp³) reductive elimination via **TS3** eventually afforded the product **3a-MgBr** and regenerated the bis-ligated nickel species Ni(L1)₂ for the next catalytic cycle. Notably, the stereoisomer transition state **TS1''** was calculated to have a higher free energy than **TS1** by 3.1 kcal/mol, which was in line with the stereochemical outcome. For comparison, DFT calculations on the catalytic cycle through mono-ligated pathway were also carried out. The results suggested that the reaction underwent similar steps as bis-ligated pathway (Figure R3-3, blue line). However, the whole potential energy surface for mono-ligated pathway was located above that of bis-ligated pathway. This results was agree with the control experiments that high loading of ligand **L1** is required for good catalytic performance and further supported the catalytic cycle in current reaction might take place through bis-L1-ligated pathway.

Comment: Computational analysis should be included to support the mechanism studies.

Response: Thanks for the reviewer's comments. According to the reviewer's comment, DFT calculations on the nickel catalyzed enantioselective methylation of aromatic C-O bond of **1a** were performed and a bis-ligated pathway was proposed (see Figure R3-2, bottom).

Comment: In the author's previous work (Fig. 1e), they use **L2** ligand for enantioselective arylation cross coupling reaction. However, only **L1** works for alkylation in similar reaction condition. The difference between **L1** and **L2** only at the connection position of alpha and beta carbon of naphthalene to NHCs. What is the reason that **L1** is more efficient than **L2**?

Response: Thanks for the reviewer's comments. To understand the failure of alkylative cross-coupling by using **L2** as the ligand, DFT calculations were performed (Figure R3-3), and found that the oxidative addition of C-O bond to nickel species Ni(L2)₂ needs to overcome a high energy barrier of 38.5 kcal/mol due to steric hindrance, which probably account for the poor performance of Ni(cod)₂/L2 catalyzed methylation of aromatic C-O bond.

Figure R3-3. Nickel catalyzed alkylative cross-coupling by using different amount of L2

Comment: What is the reason that the author used 1:2 catalyst/ligand ratio in this reaction instead of 1:1.2 ratio in previous work (Fig. 1e)?

Response: Thanks for the reviewer's comments. According to the comments, we conducted the enantioselective methylative cross-coupling of substrate **1a** by using 10 mol% Ni(cod)₂ in the presence of 12 mol% ligand **L1** (Figure R3-4), and the desired product was obtained in 96% yield with 94%, which were slightly lower than the results by using 20 mol% ligand **L1**.

Figure R3-4. Nickel catalyzed enantioselective methylation by using 12 mol% **L1**

Comment: Although the author don't explain the mechanism in this research, they proposed direct oxidative addition of Ni(0)/L to C-O bond of dinaphthylfuran in previous report (Fig. 1e). However, they modified the reaction condition by adding MgBr₂ in rational expansion on deuterated reaction, which gave a possible new mechanism for Lewis acid activated C-O activation (the mechanism in *Chin. J. Chem.* **2018**, 36, 183).

Response: Thanks for the reviewer's comments. We have read the mentioned paper carefully, and this paper have been added into the revised manuscript to explain the role of MgBr₂ in our modified conditions for cross-coupling of aromatic C-O electrophiles with CD₃-MgI. Moreover, a bis-ligated mechanism was proposed and related mechanistic studies were added in the revised manuscript.

Comment: How about dibenzofuran?

Response: Thanks for the reviewer's constructive comments. According to this comments, we prepared the substrates in Figure R3-5, and submitted them into the catalytic system, while the analysis of reaction mixtures via GC-MS revealed that no target products were delivered.

Figure R3-5. The reactivity of dibenzofuran derivatives in this cross-coupling

Comment: Typo error in the title of Fig. 1d: rout" e".

Response: Thanks for the reviewer's comments, we have corrected this typo error in the footnote of Fig. 1d.

Comment: The bond type of C-O should be consisted in all manuscript.

Response: Per the reviewer's comments, we have corrected the format of C-O bond in our revised manuscript.

Comment: CD₃-MgBr instead of MeMgBr in the footnote of Fig. 4.

Response: Thanks for the reviewer's comments, we have corrected the typo error in the footnote of Fig. 4.

Comment: Wrong compound label for compound 3s in Page S40 of supporting information.

Response: Thanks for the reviewer's comments, we have corrected the compound label in the revised supporting information.

REVIEWER COMMENTS

Reviewer #1 (Remarks to the Author):

The authors have performed control experiments and DFT calculations to study the reaction mechanism and the role of the ligand. The control experiments nicely unveiled that the bis-ligated nickel complex might serve as the catalytically active species in the alkylative cross-coupling, while mono-ligated nickel species might serve as catalytically species in arylyative cross-coupling. But I have some concerns about the DFT calculations.

(1) DFT results revealed that the bis-ligated nickel complex $\text{Ni}(\text{L1})_2$ is significantly more stable than the mono-ligated species $\text{Ni}(\text{L1})$ by 41.4 kcal/mol. This result is somehow artificial because Ni(0) prefers bidentate coordination and $\text{Ni}(\text{L1})$ only has one ligand ligated to Ni. The author should calculate $\text{Ni}(\text{L1})(\text{cod})$ instead.

(2) Same problem for the mono-ligated pathway (blue path in Fig.3). For example, Ni(II), which prefers tetra-coordinations, is tri-coordinated in $\text{TS1}'$, A' , C' , and $\text{TS3}'$. By contrast, Ni of these species is tetra-coordinated in the bi-ligated pathway. Thus, the comparison between the mono- and the bi-ligated pathway is meaningless. The author should correct the coordination states for Ni species in the mono-ligated pathway (e.g, by adding a cod ligand).

(3) DFT results suggest that L2 is not effective for the alkylation reaction due to steric hindrance. But their calculation was based on the $\text{Ni}(\text{L2})_2$ species. How about the reaction barrier with $\text{Ni}(\text{L2})(\text{cod})$?

some typos:

1. The direction of the arrow for conversion in table S1 is wrong.
2. L1 in table S2 should be L2.

Overall, the manuscript is recommended to be accepted after taking care of the points mentioned above.

Reviewer #2 (Remarks to the Author):

This revised manuscript by Cao and his co-workers has very nicely addressed all the points raised in the previous review of this manuscript. In particular, DFT calculations has been conducted to illustrate the reaction pathway involving a bis-ligated nickel complex. The origins of the stereoselectivity has been well illustrated, and the key role of ligand L1 in the alkylative cross-

coupling reactions via C–O bond activation has also been demonstrated by comparison with ligand L2 via computational studies. As such, for all of these collated improvements, this referee is in favor of acceptance of this manuscript in Nature Communications. However, my only concern is the mechanism studies regarding the higher energy barrier of the mono-ligated reaction pathway. This reviewer noticed that the authors had not considered the stabilization of intermediates or transition states by the coordination effect of cyclooctadiene ligand, which would not be reasonable according to the 18-electron rule. The authors should revise this before its publication.

Reviewer #3 (Remarks to the Author):

The authors have done a good job in replying to my comments. I gladly recommend this excellent work for publication in Nature Communication.

REVIEWER COMMENTS

Reviewer #1 (Remarks to the Author):

The authors have performed control experiments and DFT calculations to study the reaction mechanism and the role of the ligand. The control experiments nicely unveiled that the bis-ligated nickel complex might serve as the catalytically active species in the alkylative cross-coupling, while mono-ligated nickel species might serve as catalytically species in arylyative cross-coupling. But I have some concerns about the DFT calculations.

(1) DFT results revealed that the bis-ligated nickel complex $\text{Ni}(\text{L1})_2$ is significantly more stable than the mono-ligated species $\text{Ni}(\text{L1})$ by 41.4 kcal/mol. This result is somehow artificial because $\text{Ni}(0)$ prefers bidentate coordination and $\text{Ni}(\text{L1})$ only has one ligand ligated to Ni. The author should calculate $\text{Ni}(\text{L1})(\text{cod})$ instead.

(2) Same problem for the mono-ligated pathway (blue path in Fig.3). For example, $\text{Ni}(\text{II})$, which prefers tetra-coordination, is tri-coordinated in $\text{TS1}'$, A' , C' , and $\text{TS3}'$. By contrast, Ni of these species is tetra-coordinated in the bis-ligated pathway. Thus, the comparison between the mono- and the bi-ligated pathway is meaningless. The author should correct the coordination states for Ni species in the mono-ligated pathway (e.g. by adding a cod ligand).

(3) DFT results suggest that **L2** is not effective for the alkylation reaction due to steric hindrance. But their calculation was based on the $\text{Ni}(\text{L2})_2$ species. How about the reaction barrier with $\text{Ni}(\text{L2})(\text{cod})$?

some typos:

1. The direction of the arrow for conversion in table S1 is wrong.
2. **L1** in table S2 should be **L2**.

Overall, the manuscript is recommended to be accepted after taking care of the points mentioned above.

Reviewer #2 (Remarks to the Author):

This revised manuscript by Cao and his co-workers has very nicely addressed all the points raised in the previous review of this manuscript. In particular, DFT calculations has been conducted to illustrate the reaction pathway involving a bis-ligated nickel complex. The origins of the stereoselectivity has been well illustrated, and the key role of ligand L1 in the alkylative cross-coupling reactions via C–O bond activation has also been demonstrated by comparison with ligand L2 via computational studies. As such, for all of these collated improvements, *this referee is in favor of acceptance of this manuscript in Nature Communications*. However, my only concern is the mechanism studies regarding the higher energy barrier of the mono-ligated reaction pathway. This reviewer noticed that the authors had not considered the stabilization of intermediates or transition states by the coordination effect of cyclooctadiene ligand, which would not be reasonable according to the 18-electron rule. The authors should revise this before its publication.

Reviewer #3 (Remarks to the Author):

The authors have done a good job in replying to my comments. I gladly recommend this excellent work for publication in Nature Communication.

Point-by-point response to the reviewers' comments

Reviewer #1 (Remarks to the Author):

The authors have performed control experiments and DFT calculations to study the reaction mechanism and the role of the ligand. The control experiments nicely unveiled that the bis-ligated nickel complex might serve as the catalytically active species in the alkylative cross-coupling, while mono-ligated nickel species might serve as catalytically species in arylyative cross-coupling. But I have some concerns about the DFT calculations.

General response: Thanks for the reviewer's positive comments. We have conducted extensive efforts to address your insightful comments. Please see below for our detailed response.

(1) DFT results revealed that the bis-ligated nickel complex Ni(L1)₂ is significantly more stable than the mono-ligated species Ni(L1) by 41.4 kcal/mol. This result is somehow artificial because Ni(0) prefers bidentate coordination and Ni(L1) only has one ligand ligated to Ni. The author should calculate Ni(L1)(cod) instead.

Response: Thanks for the reviewer's question. According to the comment, further DFT calculations were performed. The results indicated that the mono-ligated nickel complex Ni(L1)(cod) is significantly more stable than the mono-ligated species Ni(L1) by 25.9 kcal/mol. While the bis-ligated Ni(L1)₂ is still more stable than the mono-ligated nickel complex Ni(L1)(cod) by 15.5 kcal/mol (please see the following Figure R1-1 or Fig. 3b in revised manuscript for details).

(2) Same problem for the mono-ligated pathway (blue path in Fig.3). For example, Ni(II), which prefers tetra-coordination, is tri-coordinated in TS1', A', C', and TS3'. By contrast, Ni of these species is tetra-coordinated in the bi-ligated pathway. Thus, the comparison between the mono- and the bi-ligated pathway is meaningless. The author should correct the coordination states for Ni species in the mono-ligated pathway (e.g, by adding a cod ligand).

Response: Thanks for the reviewer's constructive comments. According to the comments, DFT calculations on the mono-ligated pathway by using Ni(L1)(cod) complex was conducted. Our calculation results suggested that the oxidative addition of C–O bond to Ni(L1)(cod) via TS1'''' undergoes with a very high energy barrier ($\Delta G^\ddagger = 54.1$ kcal/mol), thus the cod-free pathway was preferred for mono-ligated process. Moreover, the similar results were also observed in the previous work by using Ni(cod)₂/NHC catalysis (e.g. Nature 2015, 524, 79–83, and J. Am. Chem. Soc. 2017, 139, 10347-10358.) (please see the following Figure R1-1 or Fig. 3b in revised manuscript for details).

Figure R1-1. PES for the overall catalytic cycle for Ni/L1 catalyzed enantioselective methylation

(3) DFT results suggest that L2 is not effective for the alkylation reaction due to steric hindrance. But their calculation was based on the Ni(L2)₂ species. How about the reaction barrier with Ni(L2)(cod)?

Response: Thanks for the reviewer's comments. According to the comment, we performed DFT calculations on the mono-ligated pathway by using Ni(L2)(cod) complex. The results suggested that the oxidative addition of C–O bond to Ni(L2)(cod) occurs via TS1^{''L2} with a very high energy barrier ($\Delta G^\ddagger = 59.8$ kcal/mol), which is much higher than that for Ni(L2)₂ via TS1^{L2} and Ni(L2) via TS1^{''L2}. (please see the following Fig. R1-2 or Fig. S2 in the revised SI for details).

Figure R1-2. PES for the overall catalytic cycle for Ni/L2 catalyzed enantioselective methylation

some typos:

1. The direction of the arrow for conversion in table S1 is wrong.

Response: Thanks for the reviewer's comments, we have corrected the error in the revised SI.

2. L1 in table S2 should be L2.

Response: Thanks for the reviewer's comments, we have corrected the typo error in the revised SI.

Overall, the manuscript is recommended to be accepted after taking care of the points mentioned above.

Response: Thank you for supporting the publication of our manuscript in Nature Communications.

Reviewer #2 (Remarks to the Author):

This revised manuscript by Cao and his co-workers has very nicely addressed all the points raised in the previous review of this manuscript. In particular, DFT calculations has been conducted to illustrate the reaction pathway involving a bis-ligated nickel complex. The origins of the stereoselectivity has been well illustrated, and the key role of ligand L1 in the alkylative cross-coupling reactions via C–O bond activation has also been demonstrated by comparison with ligand L2 via computational studies. As such, for all of these collated improvements, *this referee is in favor of acceptance of this manuscript in Nature Communications.*

General response: Thank you for supporting the publication of our manuscript in Nature Communications. We have complementally addressed your insightful comments. Please see below:

However, my only concern is the mechanism studies regarding the higher energy barrier of the mono-ligated reaction pathway. This reviewer noticed that the authors had not considered the stabilization of intermediates or transition states by the coordination effect of cyclooctadiene ligand, which would not be reasonable according to the 18-electron rule. The authors should revise this before its publication.

Response: Thanks for the reviewer's constructive comments. According to the reviewer's comments, DFT calculations on the mono-ligated pathway by using Ni(L1)(cod) complex was conducted. Our calculation results suggested that the mono-ligated nickel complex Ni(L1)(cod) is more stable than mono-ligated species Ni(L1) by 25.9 kcal/mol. However, the oxidative addition of C–O bond to Ni(L1)(cod) undergoes via TS1''' with a very high energy barrier ($\Delta G^\ddagger = 54.1$ kcal/mol), thus the cod-free pathway was preferred for mono-ligated process (See). The similar results were also observed in the previous work by using Ni(cod)₂/NHC catalysis (e.g. Nature 2015, 524, 79–83, and J. Am. Chem. Soc. 2017, 139, 10347-10358.) (please see the following Fig. R2-1 or Fig. 3b in the revised manuscript for details).

Figure R2-1. PES for the overall catalytic cycle for Ni/L1 catalyzed enantioselective methylation

Reviewer #3 (Remarks to the Author):

The authors have done a good job in replying to my comments. I gladly recommend this excellent work for publication in Nature Communication.

General response: Thank you for supporting the publication of our manuscript in Nature Communications.

REVIEWERS' COMMENTS

Reviewer #1 (Remarks to the Author):

The authors have nicely addressed all the questions listed in the previous review of this manuscript.

This reviewer is in favor to support the publication of this work on Nature Communication. But the reviewer has one little suggestion which is also related to Reviewer 2's concern. Since the currently used reference point Ni(L1)2 does not fulfill the 18-electron rule, the authors are suggested to check if Ni(L1)2(cod) is more stable than Ni(L1)2 or not, if yes, Ni(L1)2(cod) will be a more appropriate reference point.

Reviewer #2 (Remarks to the Author):

The revision has been made appropriately. I suggest the manuscript to be published in Nat. Communn.

REVIEWER COMMENTS

Reviewer #1 (Remarks to the Author):

The authors have nicely addressed all the questions listed in the previous review of this manuscript.

This reviewer is in favor to support the publication of this work on Nature Communication. But the reviewer has one little suggestion which is also related to Reviewer 2's concern. Since the currently used reference point Ni(L1)₂ does not fulfill the 18-electron rule, the authors are suggested to check if Ni(L1)₂(cod) is more stable than Ni(L1)₂ or not, if yes, Ni(L1)₂(cod) will be a more appropriate reference point.

Reviewer #2 (Remarks to the Author):

The revision has been made appropriately. *I suggest the manuscript to be published in Nat. Communn.*

Point-by-point response to the reviewers' comments

Reviewer #1 (Remarks to the Author):

The authors have nicely addressed all the questions listed in the previous review of this manuscript.

This reviewer is in favor to support the publication of this work on Nature Communication. But the reviewer has one little suggestion which is also related to Reviewer 2's concern. Since the currently used reference point Ni(L1)₂ does not fulfill the 18-electron rule, the authors are suggested to check if Ni(L1)₂(cod) is more stable than Ni(L1)₂ or not, if yes, Ni(L1)₂(cod) will be a more appropriate reference point.

Response: Thank you for supporting the publication of our manuscript in Nature Communications.

According to the comment, to identify the role of cod in the catalytic cycle, the enantioselective cross-coupling was conducted by using NiBr₂(dme) as the nickel source, and the desired product 3a was obtained in 97% yield and with 97% ee (see Table 1, entry 4 in main text or the following Figure R-1). This result indicated that cod is not essential for the cross-coupling to fulfil the catalytic cycle.

Figure R-1. The enantioselective cross-coupling by using different nickel source

Further DFT calculations were also performed. The results indicated that $\text{Ni}(\text{L1})_2(\text{cod})$ is only slightly more stable than $\text{Ni}(\text{L1})_2$ by 3.0 kcal/mol, while the dissociation of cod would occur to make space for substrate **1a** (see Figure R-2).

Thus $\text{Ni}(\text{L1})_2$ is more suitable to be used as the reference point on the basis of the results by examining different nickel source, while the $\text{Ni}(\text{L1})_2(\text{cod})$ point was added into the PES for referring.

Figure R-2. PES for the enantioselective cross-coupling by using different nickel source

Reviewer #2 (Remarks to the Author):

The revision has been made appropriately. I suggest the manuscript to be published in *Nat. Commun.*

General response: Thank you for supporting the publication of our manuscript in *Nature Communications*.